# A mixed-methods study measuring the effectiveness of a menstrual health intervention on menstrual health knowledge, perceptions and practices among young women in Zimbabwe

Mandikudza Tembo [1,2,3] Helen A Weiss [3] Leyla Sophie Larsson [2]
Tsitsi Bandason,[2] Nicol Redzo,[2] Ethel Dauya,[2] Tafadzwa Nzanza,[2]
Pauline Ishumael,[2] Nancy Gweshe,[2] Precious Ndlovu,[2] Chido Dziva Chikwari [3]
Constancia Vimbayi Mavodza,[2,4] Jenny Renju [3] Suzanna C Francis,[3]
Rashida Ferrand,[2,5] Constance R S Mackworth-Young [1,2]

For numbered affiliations see end of article.

**Correspondence to**
Dr Mandikudza Tembo;
Mandikudza.Tembo@lshtm.ac.uk

## ABSTRACT

**Objectives** While integral to women's physical and mental well-being, achieving good menstrual health (MH) remains a challenge for many women. This study investigated the effectiveness of a comprehensive MH intervention on menstrual knowledge, perceptions and practices among women aged 16–24 years in Harare, Zimbabwe.

**Design** A mixed-methods prospective cohort study with pre-post evaluation of an MH intervention.

**Setting** Two intervention clusters in Harare, Zimbabwe.

**Participants** Overall, 303 female participants were recruited, of whom 189 (62.4%) were seen at midline (median follow-up 7.0; IQR 5.8–7.7 months) and 184 (60.7%) were seen at endline (median follow-up 12.4; IQR 11.9–13.8 months). Cohort follow-up was greatly affected by COVID-19 pandemic and associated restrictions.

**Intervention** The MH intervention provided MH education and support, analgesics, and a choice of menstrual products in a community-based setting to improve MH outcomes among young women in Zimbabwe.

**Primary and secondary outcomes** Effectiveness of a comprehensive MH intervention on improving MH knowledge, perceptions, and practices among young women over time. Quantitative questionnaire data were collected at baseline, midline, and endline. At endline, thematic analysis of four focus group discussions was used to further explore participants' menstrual product use and experiences of the intervention.

**Results** At midline, more participants had correct/positive responses for MH knowledge (adjusted OR (aOR)=12.14; 95% CI: 6.8 to 21.8), perceptions (aOR=2.85; 95% CI: 1.6 to 5.1) and practices for reusable pads (aOR=4.68; 95% CI: 2.3 to 9.6) than at baseline. Results were similar comparing endline with baseline for all MH outcomes. Qualitative findings showed that sociocultural norms, stigma and taboos around menstruation, and environmental factors such as limited access to water, sanitation and hygiene

## STRENGTHS AND LIMITATIONS OF THIS STUDY

⇒ This is one of the first menstrual health studies to assess the effectiveness of a multicomponent menstrual health intervention that provided both 'hardware', through a choice of menstrual products and analgesics, and 'software', through menstrual health education and support, for young women in a community-based setting.

⇒ The study used a validated measuring tool (Menstrual Practice Needs Scale-36) that has been used in other low-income to middle-income countries to comprehensively assess the effect of the menstrual health intervention on key menstrual health outcomes, adding to a key evidence gap in robust quantitative research in menstrual health outcome measures.

⇒ The study used a mixed-methods approach that used both qualitative analysis and quantitative analysis to comprehensively assess effects of the intervention on menstrual health outcomes over time, adding to the research supporting the implementation of comprehensive evidence-based menstrual health interventions to improve poor menstrual health indicators across the globe.

⇒ Qualitative data may have been prone to social desirability bias due to the sensitive nature of the discussions around menstruation, and recall bias due to being collected retrospectively.

⇒ The conclusions of this study may be limited due to the lack of a control group, therefore other unmeasured environmental or external factors, such as natural growth and maturity, may have influenced the observed outcomes.

facilities affected the effect of the intervention on MH outcomes.

**Conclusions** The intervention improved MH knowledge, perceptions and practices among

young women in Zimbabwe, and the comprehensive nature of the intervention was key to this. MH interventions should address interpersonal, environmental and societal factors.

**Trial registration number**  NCT03719521.

## INTRODUCTION

Menstrual health (MH) encompasses a state of physical and psychosocial well-being that is informed by access to MH knowledge and products and supportive environment.[1–4] MH is integral to women's health and overall well-being,[5] but MH remains a challenge for many women, especially those in low-income and middle-income countries (LMICs).[6 7] Faced with the stigma and taboo surrounding menstruation, the lack of access to affordable menstrual products and inadequate quality MH education and support, MH remains a critically unaddressed issue.[8–12] Many girls and women experience shame, anxiety and/or untreated pain during their menstruation as they are forced to use ineffective or unhygienic alternatives such as old cloth or poor-quality disposable pads to manage their menstruation. In some instances, this lack of access to MH products and/or education on how to use menstrual products properly can lead to harmful reproductive tract infections or dangerous practices such as transactional sex in effort to access better menstrual products.[13 14]

Acceptable and effective MH interventions are integral to addressing and improving women's health outcomes.[3 15] Informed by a growing understanding of the challenges that women face in managing menstruation, interventions have been implemented in different contexts, especially in LMICs.[7 16] A systematic review of MH interventions and their effect on health outcomes among women in LMICs cautiously reported that MH interventions can inform some positive changes in MH-related outcomes such as MH knowledge levels and menstrual attitudes.[7] The review highlighted several gaps and weaknesses in MH intervention research. Many MH intervention studies tend to focus on school-going girls[17] and exclusively address either access to MH 'hardware' such as menstrual products and water, sanitation and hygiene (WASH) facilities in which to manage menstruation, or access to MH 'software' such as MH education and support targeted at demystifying and destigmatising MH-related issues, but rarely both.[7 16 18] Moreover, without validated and standardised tools and measures for MH outcomes more broadly, most studies lack robust evaluations, making it difficult to reliably assess the effect of interventions on MH-related outcomes.[7 19]

Evidence-based MH interventions that address access to menstrual products, pain management and the larger systematic factors that inform menstrual experiences are needed in Zimbabwe, as elsewhere.[20] In 2019, 21.8% of women aged 15–19 years and 18.8% of women aged 20–24 years in Zimbabwe reported not participating in social activities, school or work in the last 12 months due to menstruation.[21] Another study, by Stitching Netherlandse Vrijwilligers, conducted in 203 schools in

Masvingo, Zimbabwe reported that 72% of girls in rural Zimbabwe had never used sanitary pads.[22] Many girls and women in low-income areas throughout Zimbabwe face challenges in accessing comfortable and effective MH products and have to improvise with old cloth or newspaper.[23] Many also face challenges in accessing safe and clean WASH facilities to manage their menstruation at home, school or work. Some are unable to participate in social activities or concentrate on their work or duties due to unaddressed menstrual pain or discomfort and fear of leaking in public as they are forced to use ineffective old rags, cotton wool or tissue paper to manage their menstruation.[24 25]

The aim of this study was to address this evidence gap by investigating the effectiveness of a comprehensive MH intervention integrated within a community-based sexual and reproductive health (SRH) service on MH knowledge, perceptions and practices among young women in Zimbabwe.

## METHODS

### Study design and setting

We conducted a prospective cohort study from December 2019 to August 2021, using quantitative and qualitive methods to assess MH outcomes. The study was nested within a cluster randomised trial of community-based integrated HIV and SRH intervention for young people aged 16–24 years CHIEDZA trial registration number in clinical trials.gov: NCT03719521).[26] CHIEDZA provided a package of free-of-charge SRH services that included HIV testing and treatment, condoms, risk reduction counselling, contraception and testing and treatment of sexually transmitted infections and an MH intervention (described below) in a youth-friendly environment. CHIEDZA was delivered in 12 intervention clusters in 3 provinces across Zimbabwe (Harare, Bulawayo and Mashonaland East) from April 2019 to March 2022. Within each province, the CHIEDZA services were delivered by a team of trained healthcare providers comprising two youth workers, one counsellor, two nurses and four community health workers (CHWs). Participants in the cohort study were followed up every 3 months for a year.

At baseline and 6-month and 12-month visits, participants completed a quantitative questionnaire. At the 3-month visit, participants handed-in their completed period tracking diaries and at 9 months, participants were given another period-tracking diary to be completed and handed in at the 12-month visit.

### The menstrual health intervention

The MH intervention was preceded by a pilot (from April to July 2019) in all four Harare intervention clusters.[20] Results from the pilot were used to refine and inform the scale-up of the intervention across all 12 CHIEDZA intervention clusters. The final MH intervention available to all female CHIEDZA clients, from April 2019 to March 2022, included comprehensive MH education and

support, provision of a period-tracking diary, two pairs of normal underwear, a choice between either reusable pads (AFRIpads that can be used for up to 2 years) (www.afri-pads.com) or the menstrual cup (the Butterfly Cup that can used for up to 10 years) (www.vivalily.com), as well as soap, pain management advice and monthly pain medication (a choice between paracetamol or ibuprofen). Reusable pads and menstrual cups were used in the MH intervention as these sustainable MH products were both cost-effective and environmentally friendly.

### Cohort participants

A subset of CHIEDZA female clients from the Harare intervention clusters (Budiriro and Hatcliffe) were recruited to the prospective cohort study by two research assistants (NG and PN) between December 2019 and February 2020. Equal numbers were sought in two age-strata (16–19 and 20–24 years old) until a cohort size of 318 was reached. Clients were excluded if (1) they were currently pregnant or intended to get pregnant within the next year; (2) they experienced irregular or absent bleeding within the last 3 months or (3) had any condition that precluded informed consent or made study participation unsafe. No parental consent was needed to uptake the MH intervention as national guidelines allow for those aged 16 years and older to give independent consent to accessing SRH services.

Those who consented to participate in the cohort study received two period tracking diaries and three different types of reusable menstrual products including: (1) reusable pads; (2) a menstrual cup and (3) three pairs of period pants (VivaLily period pants that can be used for up to 2 years).

Overall, the MH intervention pilot was from April to July 2019, scale-up of MH intervention across all 12 CHIEDZA intervention clusters commenced in July 2019 and recruitment of a subset of CHIEDZA clients for the MH cohort began in December 2019.

### Sample size and COVID-19 disruptions

The planned sample size of 300 cohort participants provided 90% power to detect an increase in the proportion of participants answering all MH knowledge questions correctly from 10% at baseline to 20% at 6-month visit or 12-month visit (assuming p=0.05). This calculation was based on results from a pre-post study of an MH intervention among school girls in Uganda.[27]

COVID-19 and Zimbabwe's response strategies impacted the overall CHIEDZA trial.[28 29] Unable to go to school or work and sustain their livelihoods, many people either sent their children to, or migrated to, the rural areas in search of a more affordable environment.[30] Such migration had implications on cohort follow-up and overall exposure to the MH intervention.

### Quantitative data collection and analysis

A quantitative questionnaire informed by findings from the pilot intervention[20] and Hennegan's Menstrual Practice Needs Scale (MPNS-36),[19] was used to collect data on: (1) sociodemographic information; (2) menstruation and puberty knowledge; (3) menstrual history; (4) pain management; (5) practices and perceptions during menstruation; (6) washing practices and (7) attitudes about menstruation. All questions were translated into Shona and back-translated into English with input from MT, NG and PN. The survey was piloted with a subset of 30 CHIEDZA female clients that took up the MH intervention and iteratively revised prior to cohort recruitment.[31] The full questionnaire can be found in the online supplemental materials.

The self-administered questionnaire was completed at baseline, 6-month visit and 12-month visit. To maximise participation, we reminded participants of upcoming visits via phone. We contacted participants up to three times to reschedule if participants did not attend their scheduled visit, and offered a home visit before considering the participant lost to follow-up (LTFU) if they did not attend within 4 weeks of their scheduled visit. If participants later visited CHIEDZA again, we opportunistically invited participants to complete the questionnaire.

For this paper, relevant questions were selected from the questionnaire prior to analysis to assess the effect of the intervention on MH knowledge, perceptions and practices (for reusable pads and for the menstrual cup) over time (table 1). The data were analysed using Stata V.16.1 (StatCorp, Texas, USA). Descriptive statistics were used to describe participants' self-reported sociodemographic characteristics, including age, highest level of education, religion, marital status, employment, school, source of income, household income and household population at baseline, 6-month visit and 12-month visit. We investigated whether LTFU might potentially bias findings by comparing the sociodemographic characteristics of those present at each timepoint.

At each timepoint, and for each of the MH-related outcomes: MH knowledge, perceptions, practices for reusable pad use and practices for menstrual cup use, we calculated the average score of relevant items and the proportion of participants who answered all relevant items either 'correctly' or 'positively'. We used repeated measures logistic regression to assess the pre-post change in the proportion of participants with correct/positive outcomes from baseline to 6-month visit, and baseline to 12-month visit, respectively. Similarly, we used repeated measures linear regression to assess the pre-post change in scores for continuous variables from baseline to 6-month visit, and baseline to 12-month, respectively. We investigated potential confounders by assessing the association between each sociodemographic variable with exposure (visit timepoint) and each binary MH outcome measure. The variable was considered a potential confounder if association with both the outcome and exposure had a p<0.05. If the crude OR or regression coefficient changed by >10% on adjusting for the potential confounder, the variable was considered a confounder and was retained in the model.

**Table 1**  Questionnaire questions selected to measure menstrual health (MH) outcomes in the study

| MH outcome measures | Questions | Binary scale | |
|---|---|---|---|
| MH knowledge | Changes in the body during puberty happen because of hormones. | 0=no/do not know | 1=yes |
| | Puberty continues throughout a girl's life. | 1=no | 0=yes/do not know |
| | Menstruation in girls and women is normal. | 0=no/do not know | 1=yes |
| | When a girl gets her first menstrual period, her body can carry a child. | 0=no/do not know | 1=yes |
| | Menstrual blood is caused by the breakdown of the lining of the womb. | 0=no/do not know | 1=yes |
| | It is normal to have irregular periods as a teenager. | 0=no/do not know | 1=yes |
| | Menstruation continues throughout a girl's life. | 1=no | 0=yes/do not know |
| | A period normally lasts 2 days or less. | 1=no | 0=yes/do not know |
| | Period products that are inserted into the vagina (such as tampons and the menstrual cup) affect your virginity. | 1=no | 0=yes/do not know |
| | **Highest possible score for MH knowledge** | **9** | |
| MH perceptions | I feel dirty or impure during my menstrual period. | 0=neutral/agree/strongly agree | 1=disagree/strongly disagree |
| | I feel like I can talk to friends about menstruation. | 1=agree/strongly agree | 0=neutral/disagree/strongly disagree |
| | I feel like I can talk to my family members about menstruation. | 1=agree/strongly agree | 0=neutral/disagree/strongly disagree |
| | It is important that I keep my period secret for everyone. | 0=neutral/agree/strongly agree | 1=disagree/strongly disagree |
| | I worry that boys will tease me about my period. | 0=neutral/agree/strongly agree | 1=disagree/strongly disagree |
| | I am anxious about my next period. | 0=neutral/agree/strongly agree | 1=disagree/strongly disagree |
| | **Highest possible score for MH perceptions** | **6** | |
| MH practices (reusable pads) | I had enough water to soak or wash my menstrual products. | 1=agree/strongly agree | 0=neutral/disagree/strongly disagree |
| | I was able to wash my menstrual products when I wanted to. | 1=agree/strongly agree | 0=neutral/disagree/strongly disagree |
| | I had enough soap to wash my menstrual products when I wanted to. | 1=agree/strongly agree | 0=neutral/disagree/strongly disagree |
| | I was able to dry my products when I wanted to. | 1=agree/strongly agree | 0=neutral/disagree/strongly disagree |
| | I was worried that someone would see me while I was washing my menstrual product. | 0=neutral/agree/strongly agree | 1=disagree/strongly disagree |
| | I was worried that my menstrual products would not dry when I needed them. | 0=neutral/agree/strongly agree | 1=disagree/strongly disagree |
| | I was worried that others would see my menstrual products while they were drying. | 0=neutral/agree/strongly agree | 1=disagree/strongly disagree |
| | **Highest possible score for MH practices (reusable pads)** | **7** | |
| MH practices (menstrual cups) | I was able to sterilise my menstrual cup when I wanted after my period. | 1=agree/strongly agree | 0=neutral/disagree/strongly disagree |
| | I was able to rinse my menstrual cup when I wanted. | 1=agree/strongly agree | 0=neutral/disagree/strongly disagree |
| | I had enough water to rinse my menstrual cup. | 1=agree/strongly agree | 0=neutral/disagree/strongly disagree |
| | I was worried that someone would see me while I was rinsing my menstrual cup. | 0=neutral/agree/strongly agree | 1=disagree/strongly disagree |
| | I was worried that someone would see me while I was sterilising my menstrual cup. | 0=neutral/agree/strongly agree | 1=disagree/strongly disagree |
| | **Highest possible score for MH practices (menstrual cups)** | **5** | |

## Qualitative data collection and analysis

Following the 12-month visits, after exposure to the intervention, four focus group discussions (FGDs) were conducted with a total of 48 cohort participants in two of the intervention clusters. FGD participants were purposively selected to include a range of ages and MH product choices. A total of 12–15 young women participated in each of 4 FGDs. In each of the two clusters, one FGD was conducted with those aged 16–19 years, and one was conducted with those aged 20–24 years.

Semi-structured topic guides were informed by findings from the MH pilot study and the preliminary analysis of the quantitative data. The topic guide explored participants' perspectives on the MH intervention, what they had learnt and/or gained from the intervention, which MH products participants chose, if and how their experiences of managing menstruation had changed over time, how they felt about MH in general and how COVID-19 had affected their lives and menstrual practices. All FGDs were conducted face-to-face by experienced female qualitative researchers (MT, TN and PI) independent from the implementation team. FGDs were conducted in either Shona or English (as preferred by the participants) and took 60–75 min. Written informed consent was obtained before the FGDs were initiated and pseudonyms were used throughout for confidentiality and maintain anonymity. National guidelines indicate that parental consent is required for those under 18 years old to participate in research but due to the risk of desirability bias affecting responses and the minimal risk associated with participation in FGDs and in-depth interviews, a waiver for parental consent for participants aged 16–18 years was obtained from the ethics review bodies. FGDs were audio recorded, transcribed verbatim and then translated into English where necessary.

Analysis was guided by Braun and Clarke's approach to thematic analysis,[32] using a mix of deductive and inductive themes.[33] All transcripts were read for familiarisation and manually coded before importing them to NVivo V.12. Initial deductive codes were based on the preliminary findings from the quantitative questionnaire, and included menstrual product choice and use, MH knowledge, attitudes around MH and MH practices were based on preliminary findings from the quantitative questionnaire. Additional inductive codes were generated through transcript familiarisation, initial manual coding and analytical discussions between MT and CRSM-Y, and adapted through the coding process. These inductive codes included 'sociocultural norms', 'stigma and taboo' and 'myths'. Deductive and inductive codes were merged to develop a coding framework, which was used to code all transcripts.[34] Verbatim quotes were captured.

Qualitative analysis explored explanations and description about topics from the quantitative analysis. All the data were used to an provide in-depth understanding of and explanation on the effect of the MH intervention on MH knowledge, perceptions and practices among cohort participants.

### Patient and public involvement

This study included public involvement in the development of the MH intervention. The details of the co-development of CHIEDZA and the MH intervention nested within it have been previously described.[35 36] Briefly, we used FGDs and participatory workshops with young women aged 16–24 years and other relevant stakeholders in the community including CHWs, local community-based organisations and the Ministry of Health and Child Care to investigate MH gaps in the community, develop the MH intervention and inform the research questions for this study.

To identify the most relevant MH gaps and needs in our intervention communities, we worked with key stakeholders to inform a Theory of Change (ToC) for the MH intervention. The ToC articulated the critical pathways to improve key MH outcomes among young women in Zimbabwe and guided the development of the MH intervention components. The intervention was then piloted from April to June 2019 and feedback from both healthcare providers and clients were used to refine and inform the scaled-up implementation of the MH intervention.

Overall, patients and/or the public were involved in the research process, design of the study, the recruitment and conduct of the study, or the dissemination of study results.

### RESULTS
#### Participant characteristics

Three hundred eighteen female clients were screened at two of the CHIEDZA clusters (Hatcliffe and Budiriro) between December 2019 and February 2020. Of those screened, two were outside the eligible age range and 13 (4.1%) declined to participate due to time constraints. A total of 303 participants were therefore enrolled in the cohort study (figure 1).

Sociodemographic characteristics of participants at each timepoint are presented in table 2. Of the 303 participants at baseline, half (51.2%) were aged 16–19 years and half (51.8%) had completed O-level education (equivalent to the UK GCSE). One-third (35.0%) of participants were in school and 87.8% did not have a job. Half (51.2%) obtained money or income from a relative and 28.2% obtained money from a partner; 61.1% of participants had never married and all came from low-income or moderate-income households. Of the 303 participants that were seen at baseline, 189 (62.4%) were seen at 6-month visits and 184 (60.7%) at 12-month visits. Overall, cohort participant characteristics were broadly similar at baseline, 6-month visits and 12-month visits.

#### MH knowledge, perceptions and practices

For MH knowledge, perceptions and MH practices for reusable pads, the mean scores and the proportion of participants with all correct or positive responses increased from baseline to 6-month visit and decreased from 6-month visit to 12-month visit (table 3). For MH practices for menstrual cups, the mean score and proportion with all positive responses increased from baseline to 6-month visit and from 6-month visit to 12-month visit. For MH knowledge and perceptions, there was strong evidence of an increase in mean scores from baseline to 6-month visit with MH knowledge mean score reflecting the highest improvement (1.62; 95% CI: 1.27 to 1.97) over time. There was also an increase in mean score from baseline to 12-month visit for MH perceptions.

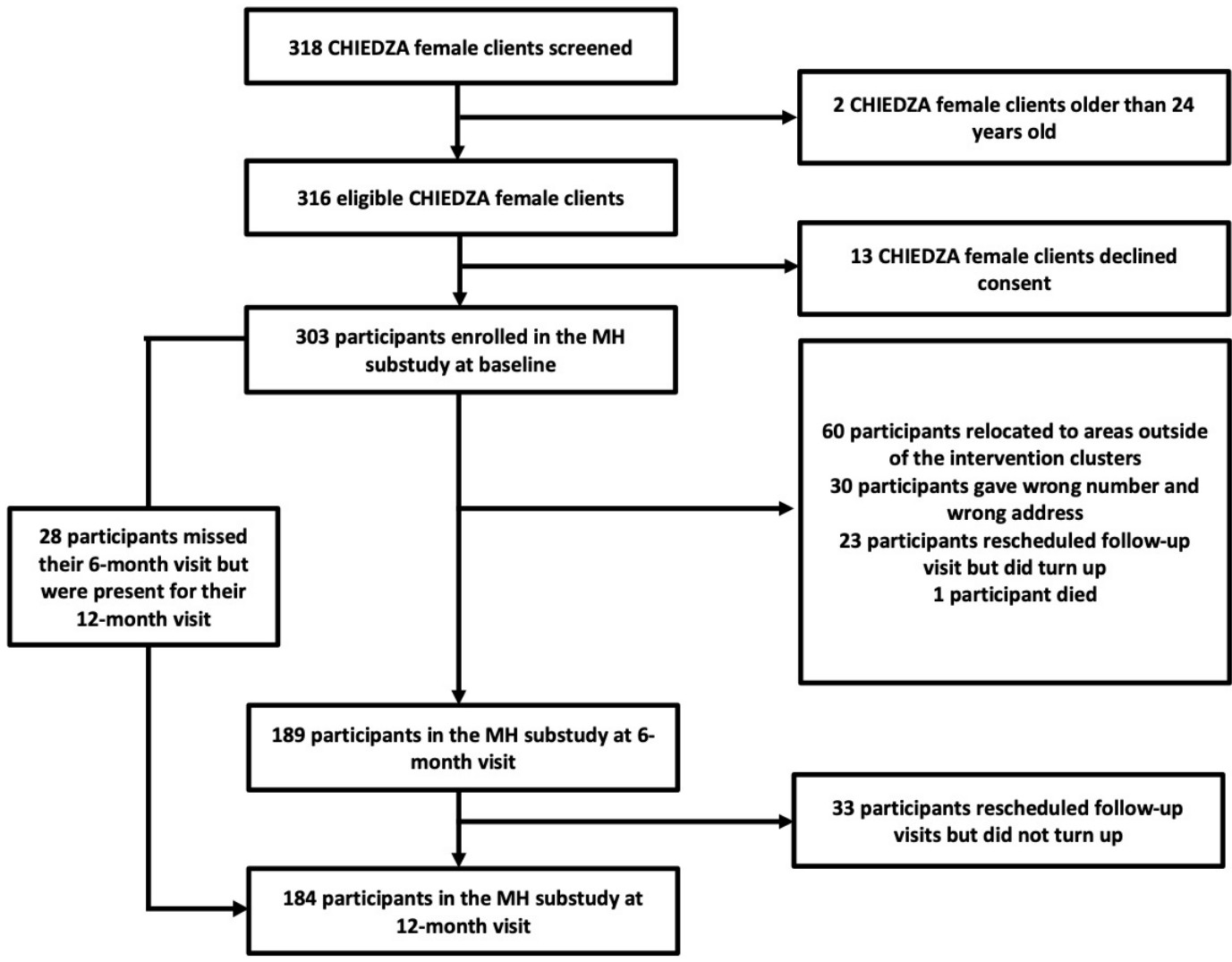

**Figure 1 Screening, recruitment, and follow-up flow chart for MH cohort substudy**

**Figure 1** Screening, recruitment and follow-up flow chart for menstrual health (MH) cohort study.

After adjusting for confounders (age, level of education and household population), there was strong evidence of an increase in MH knowledge, perceptions and practices over time (table 3). At 6-month visit, participants were more likely to respond correctly or positively for all items for MH knowledge adjusted OR (aOR)=12.14; 95% CI: 6.8 to 21.8), MH perceptions (aOR=2.85; 95% CI: 1.6 to 5.1) and practices for reusable pads (aOR=4.68; 95% CI: 2.3 to 9.6) than at baseline. Similarly, there was strong evidence of an improved response on MH knowledge (aOR=7.8; 95% CI: 4.3 to 14.2), MH perceptions (aOR=2.53; 95% CI: 1.4 to 4.6) and MH practices (reusable pads) (aOR=3.64; 95% CI: 1.8 to 7.5) from baseline to 12-month visit. However, there was a slight decrease in odds of improved responses between 6-month and 12-month visits for MH knowledge, perceptions and practices for reusable pads.

Qualitative findings provided details and explanations to the increases in knowledge, perceptions and practices. They explained how an unmet need for MH education, support and products facilitated uptake of the MH intervention:

> All I want to say is CHIEDZA has helped me so much. For me, I no longer have stress to include pads on my grocery list. (Hatcliffe, FGD, 20–24 years old)

### MH knowledge

Participants reported acquiring MH knowledge through the intervention. When asked questions about menstruation in the FGDs after the intervention, menstrual pain management and how to use, wash, dry and store the reusable menstrual products given to them through the MH intervention, almost all of the participants responded confidently and accurately. For example, when asked about use of the menstrual cup and 'virginity' they responded:

**Table 2** Sociodemographic characteristics and menstrual health (MH) outcomes at each time point

| | Sociodemographic characteristics of MH cohort participants | | | | | | | |
| --- | --- | --- | --- | --- | --- | --- | --- | --- |
| | Baseline (n=303) | | 6-month visit (n=189) | | | 12-month visit (n=184) | | |
| | N | % | N | % | P value | N | % | P value |
| Age (years) | | | | | | | | |
| 16–19 | 155 | 51.2% | 84 | 54.3% | 0.169 | 68 | 53.3% | 0.002 |
| 20–24 | 148 | 48.8% | 105 | 45.7% | | 116 | 46.7% | |
| Highest level of education | | | | | | | | |
| Less than primary school or primary school completed | 110 | 36.3% | 66 | 34.9% | 0.001 | 55 | 29.9% | 0.178 |
| O-levels completed | 157 | 51.8% | 90 | 47.6% | | 97 | 52.7% | |
| More than O-levels completed | 36 | 11.9% | 33 | 17.5% | | 32 | 17.4% | |
| Currently in school | | | | | | | | |
| Yes | 106 | 35.0% | 72 | 38.1% | 0.485 | 61 | 33.2% | 0.680 |
| No | 197 | 65.0% | 117 | 61.9% | | 123 | 66.8% | |
| Paying job | | | | | | | | |
| Yes | 37 | 12.2% | 20 | 10.6% | 0.583 | 29 | 15.8% | 0.267 |
| No | 266 | 87.8% | 169 | 89.4% | | 155 | 84.2% | |
| Marital status | | | | | | | | |
| Never married | 185 | 61.1% | 126 | 66.7% | 0.271 | 114 | 61.9% | 0.971 |
| Married, or living as if married | 110 | 36.3% | 61 | 32.3% | | 64 | 34.8% | |
| Single (widowed or divorced) | 8 | 2.6% | 2 | 1.00% | | 6 | 3.3% | |
| Religion | | | | | | | | |
| Not religious | 8 | 2.6% | 4 | 2.10% | 0.808 | 3 | 1.6% | 0.677 |
| Christian | 251 | 82.8% | 154 | 81.5% | | 157 | 85.3% | |
| Other | 44 | 14.5% | 31 | 16.4% | | 24 | 13.1% | |
| Household population | | | | | | | | |
| Lives alone | 10 | 3.3% | 0 | 0.0% | 0.007 | 3 | 1.6% | 0.026 |
| Lives with 1–3 other people | 116 | 38.3% | 58 | 30.7% | | 55 | 29.9% | |
| Lives with 4–6 other people | 147 | 48.5% | 101 | 53.4% | | 93 | 50.5% | |
| Lives with >6 other people | 30 | 9.9% | 30 | 15.9% | | 33 | 17.9% | |
| Total household income | | | | | | | | |
| Low-income | 149 | 49.2% | 93 | 49.2% | 0.995 | 84 | 45.7% | 0.451 |
| Moderate-income | 154 | 50.8% | 96 | 50.8% | | 100 | 54.3% | |
| Source of income* | | | | | | | | |
| I do not get money from anyone | 59 | 19.6% | 28 | 15.0% | 0.525 | 35 | 19.4% | 0.998 |
| I get money from a relative | 154 | 51.2% | 105 | 56.5% | | 91 | 50.6% | |
| I get money from my partner | 85 | 28.2% | 52 | 28.0% | | 52 | 28.9% | |
| I get money from somewhere else | 3 | 1.0% | 1 | 0.5% | | 2 | 1.1% | |

*Missing data for 'source of income' sociodemographic characteristics at baseline (n=303), 6-month visit (n=189) and 12-month visit (n=184).

They (the CHIEDZA healthcare providers) explained… They said that it's not bad and it does not affect someone's virginity. I believe that it doesn't affect virginity. (Budiriro, FGD, 16–19 years old)

Their responses evidenced that participants had both retained the MH education from the intervention and gained additional MH knowledge on 'best practices' based on their experiences using the products. When

asked about washing and drying the reusable pads, participants responded:

Don't wear [the pads] when damp… That's when it smells bad. It must dry properly because it's made of cotton… (Budiriro, FGD, 16–19 years old)

While most of the conversation around MH knowledge revealed accurate information retention, some of the responses around menstrual cups and pain management highlighted how sociocultural norms and persistent

**Table 3** MH outcome mean scores (M) and ORs for the proportion answering all items correctly over time for participants at 6-month visit versus baseline, and 12-month visit versus baseline, respectively

| MH outcome | Baseline (n=303) | | 6-month visit (n=189) | | | | | 12-month visit (n=186) | | | | |
|---|---|---|---|---|---|---|---|---|---|---|---|---|
| | M | % with all correct/positive responses | M | % with all correct/positive responses | Absolute increase in mean score (95% CI) | aOR (95% CI) | P value | M | % with all correct/positive responses | Absolute increase in mean score (95% CI) | aOR (95% CI) | P value |
| Knowledge | 5.81 | 5.3% | 7.45 | 42.3% | 1.62 (1.27 to 1.97) | 12.14 (6.77 to 21.76) | <0.001 | 6.92 | 32.6% | 1.08 (0.73 to 1.42) | 7.80 (4.3 to 14.16) | <0.001 |
| Perceptions | 2.54 | 9.2% | 3.03 | 21.2% | 0.49 (0.23 to 0.75) | 2.85 (1.59 to 5.09) | <0.001 | 2.90 | 19.0% | 0.36 (0.10 to 0.62) | 2.53 (1.40 to 4.55) | <0.005 |
| Practices (reusable pads) | 4.41 | 14.2% | 5.58 | 40.6% | 1.17 (0.82 to 1.52) | 4.68 (2.28 to 9.58) | <0.001 | 5.53 | 34.4% | 0.96 (0.60 to 1.30) | 3.64 (1.77 to 7.48) | <0.001 |
| Practices (menstrual cups) | 3.08 | 21.1% | 3.80 | 40.6% | 0.55 (0.40 to 1.15) | 39.9 (0.05 to 34638.94) | <0.3 | 4.00 | 48.5% | 0.95 (0.36 to 1.53) | 1200.42 (0.44 to 3 297 620.00) | <0.1 |

*MH knowledge and MH practices (reusable pads) ORs adjusted for age and household population, MH perceptions OR adjusted for level of education and MH practices (menstrual cups) OR adjusted for age.
aOR, adjusted OR.

myths overrode MH education in the intervention. For example, when asked about menstrual pain, one participant said:

> For me, when I was a girl, I didn't experience period pain. It came after birth. I think I got it from my hospital bed. I think the bed I used was used by someone who had period pain. (Budiriro, FGD, 20–24 years old)

## MH perceptions

After the intervention, participants reported feeling more confident about knowing how to manage their menstruation using the reusable menstrual products. Many participants also reported feeling '*proud*' and more positive about their menstruation. They described the MH intervention as reducing the shame they felt about menstruation. Some participants stated that they were determined to defy sociocultural norms that framed menstruation as a shameful 'personal secret' no one else should know about:

> As for me, at our homestead there are boys but it's that, I dry my things [reusable pads] outside on the washing line because those boys they know about menstruation. They will marry their wives and they will also go on their monthly periods, even their mothers who gave birth to them, they go on their periods, it's not something new…" (Budiriro, FGD, 16–19 years old)

Despite the MH education and support, however, some still faced stigma and expressed feeling shame and anxiety around menstruation. Fear of 'smelling like fish' or leaking menstrual blood persisted and informed MH product choice and practices. This shame around menstruation prevented them from feeling comfortable and confident during menstruation and when using and/or washing their menstrual products of choice:

> I thought it better to dry it [my pads] inside the house because people would want to know, what that is. They would also know about my period, when it started… so I will not feel alright. (Hatcliffe, FGD, 20–24 years old)

For some, longstanding sociocultural norms around menstruation overrode the positive menstrual perceptions gained through the intervention, and impacted MH practices, despite participants' gained knowledge.

> I stay with my aunt, my husband's sister. So, when I got the cup, she asked me if she can use the cup… The cup is not meant for girls, girls should use pads. (Hatcliffe, FGD, 20–24 years old)

## MH practices

Participants were excited to have the opportunity to choose and use different types of menstrual products through the intervention. Product choice and menstrual

practice was informed both by the knowledge gained through the intervention, as well as external sociocultural factors. Sociocultural beliefs around the menstrual cup causing the loss of 'virginity' prevented many participants from using the cup, leading to low cup uptake. This was despite the issues of cup use and virginity being discussed during MH education sessions embedded in the MH intervention.

> Youths, they fear, and I'm one of them. I never attempted to use it because of fear, I was afraid to lose virginity, but I heard that if you use it, it's alright and you don't feel anything, everything will be normal. (Budiriro, FGD, 16–19 years old)

Since products were used away from the CHIEDZA intervention site, at home, the influence of sociocultural factors was strong.

> For me, I remember when we got these products, I was excited. When I got home, I showed my mom and she just said, 'I don't like that thing [the cup]'. So, you even become afraid of a thing which your mom does not approve of. (Budiriro, FGD, 20–24 years old)

While most participants chose not to use the menstrual cup, those that did, reported positive user experiences that encouraged them to continue cup use over time:

> I was scared at first to use the cup but when I started using it [chuckles] It is very rare to see my pads on the washing line. (Hatcliffe, FGD, 20–24 years old)

Environmental factors such as location (at home vs at work or school) and access to soap, water and/or facilities to safely wash and dry reusable products also informed MH practices. These factors linked to the aforementioned perceptions of shame and anxiety around menstrual blood, with cleanliness felt to be a priority.

> There is a difference, if I'm at home and school, they are two different things. I don't use period pants when I'm at school because it does not handle blood for long time and the thing is you will be coming in and out in classroom… My colleagues will become suspicious. So, if I'm home I use period pants and when I'm school I use reusable pads. (Budiriro, FGD, 16–19 years old)

> When travelling I prefer disposable pads because you never know where you go, you might not have water to use, and these things they need privacy, because when washing or drying them everyone there would be curious to know about it. (Budiriro, FGD, 20–24 years old)

For some, menstrual flow also informed MH practices, including using more than one menstrual product at a time.

> I can say first three days will be using a combination period pants and reusable pad. But last day will just use period pant. (Budiriro, FGD, 16–19 years old)

Additional quotes from the FGDs are highlighted in table 4.

## DISCUSSION

This study provides evidence of improvements in key MH outcomes of knowledge, perceptions and practices, through an intervention providing young women with MH products, education and support in a community-based setting in urban Zimbabwe. The MH intervention led to knowledge acquisition on menstruation, menstrual pain management and menstrual product use; increased confidence in menstrual management and improved MH practices. Sociocultural and environmental factors, such as myths and taboos around menstruation, cultural beliefs and access to WASH facilities at home, school or work, also informed the MH outcomes, but were outside the intervention's scope, and may have attenuated the positive effects of the intervention over time.

Our baseline findings add to evidence highlighting poor MH knowledge, perceptions and practices, among young women in LMICs.[6 21] With MH shrouded in secrecy and taboo, we supplement evidence that young women often reach menarche with limited information, and sometimes misinformation, compounded by a lack of access to MH pain relief and products of choice, leading to poor experiences of managing menstruation.[10 20 37] We therefore emphasise the need for comprehensive MH interventions and provide evidence of an intervention to improve key MH outcomes.

The MH outcomes—knowledge, perceptions and practices—were closely connected, highlighting the importance of integrated interventions targeting all three. Increased MH knowledge led to better MH practices and perceptions and better MH practices and perceptions led to better MH experiences overall. The value of offering choices of MH product enhanced MH experiences as young women could choose the product that best suited their circumstance and led to increased experiential knowledge on how to use, wash, dry and store different products in young women's own specific environments. When encouraged to, participants were able to share their positive menstrual experiences and transfer that gained knowledge to others: changes in MH practices and perceptions thereby enabled MH information exchange and support.[38] This process of acquiring and transferring MH knowledge and sharing menstrual experiences is often absent in most communities, without much needed MH discussion, correct information dissemination and MH-related healthcare seeking behaviours.[39–41]

The study highlights the influence of both the internal intervention components as well as external community social norms and access to WASH facilities at work, home or school, as they interacted to inform young women's

**Table 4** Summary of quotes from FGD participants

| Main themes | Quotes from the FGD |
|---|---|
| MH knowledge | "When I clean them [reusable pads], I fetch water in a bucket, go to the bathroom, wash using green bar soap, rinse and put them on the washing line to dry, then put them in a bag and then in my drawer". (Hatcliffe, FGD, 20–24 years old)<br>"If you give birth, period pain goes away". (Hatcliffe, FGD, 20–24 years old)<br>"Don't wash them using detergents like washing powder, you should use green bar the dry them in sunlight for it to dry then you can you it again". (Hatcliffe, FGD, 16–19 years old)<br>"We were told to boil the cup. It was said to destroy germs plus I think it can soften the hardness of it". (Budiriro, FGD, 20–24 years old) |
| MH perceptions | "When I hear about CHIEDZA, I feel happy because I think of the products they provide for the girls in my community". (Hatcliffe, FGD, 16–19 years old)<br>"I have difficulties when drying, at our homestead there are boys, so I dry my products inside our house…" (Budiriro, FGD, 16–19 years old)<br>"As for my mom, she said 'don't use that thing… Just wash other pads that you have'. She didn't want too much [more than that]. (Budiriro, FGD, 16–19 years old)<br>"The disadvantages are people are not liking the cup especially girls they say that it makes then lose their virginity and some when they wear it, they feel like they are having sexual intercourse when walking of the tail". (Hatcliffe, FGD, 20–24 years old) |
| MH practices | "I can say girls with 16–19 years are the ones who like reusable pads while the 20–24 they need cup". (Budiriro, FGD, 16–19 years old)<br>"On my first days I use reusables then on my last days I used period pants because the blood flow will be minimal…" (Hatcliffe, FGD, 16–19 years old)<br>"[I wear] period pant first day of period because I will be having a light flow but other days, I will be using pads". (Budiriro, FGD, 16–19 years old)<br>"When travelling I prefer disposable pads because you never know where you go, you might not have water to use, and these things they need privacy, because when drying them everyone there would be curious to know about it". (Budiriro, FGD, 20–24 years old)<br>"Whenever I feel period pain, I take a pill". (Budiriro, FGD, 20–24 years old) |

FGD, focus group discussions.

menstrual experiences and MH outcome measures.[19] This understanding of the multifaceted nature of MH is consistent with the growing understanding that MH plays a key role in the lived experiences of women across the globe. Comprehensive MH interventions must thus address biological, personal, interpersonal, environmental and societal MH-related gaps and barriers.[5]

The decrease in all three outcomes from 6-month to 12-month visits although small, underscores the need for interventions that address social norms which can otherwise impede long-term MH improvements. Previous studies have similarly had limited impact on social norms around menstruation. In a study looking at the effect of a community-based MH intervention for adolescent girls in India,[42] results showed a significant improvement in MH practices, but only a marginal decline in the social and religious restrictions that informed attitudes towards menstruation. While there may be additional factors that informed the 12-month visit dip in MH knowledge, perceptions and practices in our study, such as participants' levels of engagement with MH activities within CHIEDZA, our results suggest the need for MH interventions that include community members and prioritise community sensitisation and education around MH-related issues if there are to be acceptable, effective and sustainable over time.

This is one of the first MH studies to assess the effectiveness of a comprehensive community-based MH intervention that provided both MH 'hardware', through menstrual products, and 'software', through MH education, for young women in a community-based setting.[16] Unlike most MH interventions, our study offered participants a choice of MH products and analgesics and used a mixed-methods approach to assess the effectiveness of this comprehensive MH intervention and the factors that informed MH product choice and patterns of use over time. Unusually, our study looks comprehensively at knowledge, perceptions and practices together, and uses a validated measurement tool (MPNS-36) that has been used in other LMICs such as Uganda.[19 43]

In terms of limitations, there was substantial LTFU which may have led to bias. However, a comparison of participant sociodemographic characteristics at each visit showed no difference between those retained in the study and those LTFU. Qualitative data may have been prone to recall bias due to being collected retrospectively, and social desirability bias, due to the sensitive nature of the discussions. While observed differences before and after the study were assumed to be due to the intervention, with the lack of a control group, other external factors, such as natural growth, ageing or maturity or other environmental changes may have influenced MH outcomes.[44] While this was further explored qualitatively, temporal bias may be present. The cohort participants were recruited in two randomly selected CHIEDZA intervention clusters (Hatcliffe and Budiriro) within one province (Harare) of Zimbabwe. Therefore, the relatively small number of participants in the cohort came from similar backgrounds and have similar characteristics making the study results less generalisable.

# CONCLUSION

Overall, the study results showed that the comprehensive MH intervention was effective in improving MH knowledge levels, perceptions and practices among young women. Provision of a combination of comprehensive MH education, analgesics, a choice of MH products and support over time were key to intervention effectiveness and success. While exposure and access to the youth-friendly service provision of MH education and resources facilitated improvements in MH outcome measures, these were mediated by external and contextual factors such as sociocultural norms and environmental conditions. It is important that MH gaps and barriers are tackled using a holistic approach that frames MH as a health and human rights issue and engages with both individuals and the wider communities in which they operate.

## Author affiliations

[1]Department of Global Health and Development, London School of Hygiene and Tropical Medicine Faculty of Public Health and Policy, London, UK
[2]Biomedical Research and Training Institute, Harare, Zimbabwe
[3]MRC International Statistics & Epidemiology Group, Department of Infectious Disease Epidemiology, London School of Hygiene & Tropical Medicine, London, UK
[4]Department of Public Health, Environments and Society, London School of Hygiene & Tropical Medicine, London, UK
[5]Clinical Research Department, London School of Hygiene and Tropical Medicine Department of Clinical Research, London, UK

**Acknowledgements** The authors would like to thank all the participants in the study.

**Contributors** MT is the guarantor and designed the menstrual health intervention, and collected, analysed and interpreted the quantitative and qualitative data, and drafted the manuscript. CRSM-Y assisted in analysing the qualitative data and was a major contributor in writing the manuscript. HAW assisted in analysing the quantitative data and was a major contributor in writing the manuscript. LSL assisted in analysing the quantitative data. NG, PN, TN and PI assisted in the collection of the qualitative and quantitative data. TB and NR assisted with quantitative data collection and management. RF, CDC, CVM, and ED contributed to the coordination of the study and critical revisions of the manuscript. SCF and JR contributed to the design of the menstrual health intervention and the analysis plan. RF is the Principal Investigator of the CHIEDZA trial. All authors read and approved the final manuscript.

**Funding** Swiss Agency for Development and Cooperation funding, the Wellcome Trust through a Senior Fellowship to RAF (206316/Z/17/Z) and the Fogarty International Centre of the National Institutes of Health under Award Number D43 TW009539.

**Competing interests** None declared.

**Patient and public involvement** Patients and/or the public were involved in the design, or conduct, or reporting, or dissemination plans of this research. Refer to the 'Methods' section for further details.

**Patient consent for publication** Not applicable.

**Ethics approval** This study was approved by the Medical Research Council of Zimbabwe (MRCZ/A/2387), the London School of Hygiene and Tropical Medicine ethics committee (16124/RR/11602) and the Biomedical Research and Training Institute Institutional Review Board (AP149/2018). Participants gave informed consent to participate in the study before taking part.

**Provenance and peer review** Not commissioned; externally peer reviewed.

**Data availability statement** Data are available on reasonable request. The datasets used and/or analysed during the current study are available from the corresponding author on request.

**ORCID iDs**
Mandikudza Tembo http://orcid.org/0000-0002-4520-3317
Helen A Weiss http://orcid.org/0000-0003-3547-7936
Leyla Sophie Larsson http://orcid.org/0000-0003-4869-4630
Chido Dziva Chikwari http://orcid.org/0000-0003-1617-3603
Jenny Renju http://orcid.org/0000-0001-5650-1902
Constance R S Mackworth-Young http://orcid.org/0000-0002-9725-7931

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
