## [Reviewer comments · BMJ Open]

ARTICLE DETAILS

TITLE (PROVISIONAL)	A mixed-methods study measuring the effectiveness of a menstrual health intervention on menstrual health knowledge, perceptions, and practices among young women in Zimbabwe
AUTHORS	Tembo, Mandikudza; Weiss, Helen; Larsson, Leyla; Bandason, Tsitsi; Redzo, Nicol; Dauya, E; Nzanza, Tafadzwa; Ishumael, Pauline; Gweshe, Nancy; Ndlovu, Precious; Dziva Chikwari, Chido; Mavodza, Constancia; Renju, Jenny; Francis, Suzanna; Ferrand, Rashida; Mackworth-Young, Constance

VERSION 1 – REVIEW

REVIEWER	AMOAKOH-COLEMAN, MARY University of Ghana
REVIEW RETURNED	22-Nov-2022

GENERAL COMMENTS	MH Interventions and Outcomes  1. Study Design: The study design as described through the paper is Mixed Methods. Under Quantitative, a prospective cohort study, nested within a cluster RCT was carried out. Please revise to clearly reflect this. Currently it is erroneously described. 2. Use of MH Products: It is not clear from the Background, the extent of use of MH products in Zimbabwe which necessitate this intervention. What proportion of women (before the study) had access to reusable pads and menstrual cups? Why were disposable pads not considered in this study? 3. MH Intervention: The supplies of soap, pain management advise, and pain medication was for how long? Was it for 12 months? Please indicate clearly. 4. Cohort Participants: What do you mean by clients who chose to take up any component of the MH Intervention? Does it mean the 303, 189 and 184 participants at baseline, six and 12 months had different MH intervention and thus the intervention was not uniform for all participants? This has implications on the results. Please clarify and explain why you chose this route? 5. MH Outcome: the study looks at MH Knowledge, perceptions, practice of use of reusable pads and practice of use of menstrual cups. Which of these if the main outcome, and which of them was estimated as 10% at baseline in calculating the sample size? 6. Why were women 12-15 years of age included in the FGDs when they were not part of the Intervention group? What was the reason for including them? Did they not dilute the wealth of information? 7. Table 2: You mention that the effect of LTFU is examined, and characteristics were similar at the three study points. However, the
--

	Table does not show any test of difference or association to prove that point. Kindly address that so we can be sure the characteristics are truly similar. 8. Table 2: For the numbers for source of income, does it include those who have a job? Because the options do not reflect that. 9. Table 3: Practices (menstrual cups) has very high Confidence Intervals for the measures of effect. What is accounting for that? And how do you interpret this? 10. Discussion: You could discuss the possibility of lack of repeated interventions ..eg for the education, on the reduction in the measure of outcomes over time. Again under limitations, you have not yet shown that LTFU had no effect so please revise.
--	--

REVIEWER	Farokhi, Moshtagi University of Texas Health Science Center at San Antonio Libraries
REVIEW RETURNED	30-Nov-2022

GENERAL COMMENTS	The manuscript is well written. There are a few recommendations that can enhance the manuscript's quality. Regarding the cultural and religious barriers or influences for MH (pages 405-408) authors include sociocultural and environmental factors, including cultural beliefs, as factors outside the intervention's scope, that may attenuate positive effects of the intervention over time. Could the cultural and religious influences be written under the exclusion or exclusion criteria as well? Page 7 Lines 147-149 Regarding the fact that the study was nested within a cluster randomized trial of community-based integrated HIV and SRH intervention for young people aged 16-24 years (CHIEDZA); this information should also be reported as a bias/limitation since only a select or convenience sample of females (CHIEDZA clients) who attend these sites participated. Great use of repeated measures linear regression to assess the pre-post surveys. Pages 19-22, under results, it may be redundant to use quotes in the main manuscript as well presenting the exact quotes as Table 4 on page 23.
---

VERSION 1 – AUTHOR RESPONSE

	Reviewer Comments	Author Responses
Reviewer 1		
	Thank you for reporting this interesting material.	Thank you for your review of this paper.
1.	Study Design: The study design as described through the paper is Mixed Methods. Under Quantitative, a prospective cohort study, nested within a cluster RCT was carried out. Please revise to clearly reflect this. Currently it is erroneously	Thank you for your comment. The study used data from a prospective cohort nested within a cluster randomized trial called CHIEDZA. We also used quantitative and qualitative methods, We

	described.	have revised the manuscript to make this more clear: “Design: A mixed-methods prospective cohort study with pre-post evaluation of an MH intervention.” Page 3, Line 53
2.	Use of MH Products: It is not clear from the Background, the extent of use of MH products in Zimbabwe which necessitate this intervention. What proportion of women (before the study) had access to reusable pads and menstrual cups? Why were disposable pads not considered in this study	There is limited access to reusable and disposable MH products in Zimbabwe. We have included additional text to the revised manuscript to reflect this: “Another study, by Stitching Netherlandse Vrijwilligers (SNV), conducted in 203 schools in Masvingo, Zimbabwe reported that 72% of girls in rural Zimbabwe had never used sanitary pads (22). Many girls and women in low-income areas throughout Zimbabwe face challenges in accessing comfortable and effective MH products and have to improvise with old cloth or newspaper (23). Many also face challenges in accessing safe and clean WASH facilities to manage their menstruation at home, school, or work.” Page 7, Lines 138 – 142 We did not consider disposable pads for the following reasons:  - We wanted to provide sustainable menstrual products participants could continue to use even after the intervention had ended. - It would have been very difficult to calculate and distribute the appropriate number of pads to be given to participants every month - Given the huge need for MH products in the communities, it would have been impossible to ensure that participants were using the pads and not selling or sharing pads with others in the community

		- Lack of disposal facilities in the communities made disposable pads less environmentally friendly We have added additional text in the revised manuscript to summarize this: “Reusable pads and menstrual cups were used as these sustainable MH products were both cost-effective and environmentally-friendly.” Page 8, Lines 178-9
3.	MH Intervention: The supplies of soap, pain management advise, and pain medication was for how long? Was it for 12 months? Please indicate clearly.	The MH intervention was available to participants throughout the intervention period from April 2019 – March 2022. We have revised the manuscript to reflect this: “The final MH intervention available to all female CHIEDZA clients, from April 2019 to March 2022, included comprehensive MH education and support, provision of a period-tracking diary, two pairs of normal underwear, a choice between either reusable pads (AFRIpads that can be used for up to two years) (www.afripads.com) or the menstrual cup (the Butterfly Cup that can used for up to ten years) (www.vivalily.com), as well as soap, pain management advice and monthly pain medication (a choice between paracetamol or ibuprofen).” Page 8, Line 169
4.	Cohort Participants: What do you mean by clients who chose to take up any component of the MH Intervention? Does it mean the 303, 189 and 184 participants at baseline, six and 12 months had different MH intervention and thus the intervention was not uniform for all participants? This has implications on the results. Please clarify and explain why you chose this route?	Thank you for your comment. We agree the methods you describe would have implications on the results and the way this was written up in the manuscript was unclear. Cohort participants were recruited from female CHIEDZA clients that were

		interested in taking up any of the MH intervention components i.e. pain management advice, medication, general MH education or an MH product. If participants agreed to be a part of the cohort study, they then received the full MH intervention. This means those at baseline, midline, and endline all received the same MH intervention. We have revised the manuscript to make this clear: “A subset of CHIEDZA female clients from the Harare intervention clusters (Budiro and Hatcliffe) were recruited to the prospective cohort study by two research assistants (RAs) (NG and PN) between December 2019 to February 2020.” Page 9, Line 182 “Those who consented to participate in the cohort study received two period tracking diaries and three different types of reusable menstrual products including: 1) reusable pads; 2) a menstrual cup; and 3) three pairs of period pants (VivaLily period pants that can be used for up to two years).” Page 9, Line 191 - 193
5.	MH Outcome: the study looks at MH Knowledge, perceptions, practice of use of reusable pads and practice of use of menstrual cups. Which of these if the main outcome, and which of them was estimated as 10% at baseline in calculating the sample size?	Thank you for your comment. The main objective of the study was to investigate the effect of the MH intervention on MH knowledge, practices, and perceptions. All three of these measures were part of the main outcome.

		The sample size of 300 participants was calculated based on MH knowledge. We have revised the manuscript to clearly articulate this: “The planned sample size of 300 cohort participants provided 90% power to detect an increase in the proportion of participants answering all MH knowledge questions correctly from 10% at baseline to 20% at six-month visit or 12-month visit (assuming p=0.05). This calculation was based on results from a pre-post study of an MH intervention among school-girls in Uganda.” Page 9, Line 198 - 200
6.	Why were women 12-15 years of age included in the FGDs when they were not part of the Intervention group? What was the reason for including them? Did they not dilute the wealth of information?	We did not include women 12 – 15 years old. We included 12 – 15 young women aged 16 – 24 years old. This is reflected in the manuscript: “12-15 young women participated in each of four FGDs. In each of the two clusters, one FGD was conducted with 16-19-year-olds, and one was conducted with 20–24-year-olds.” Page 13, Lines 251 - 253
7.	Table 2: You mention that the effect of LTFU is examined, and characteristics were similar at the three study points. However, the Table does not show any test of difference or association to prove that point. Kindly address that so we can be sure the characteristics are truly similar.	Thank you for your comment. We have included the p values in Table 2 to reflect the results of the statistical analysis used to assess differences between cohort participants as baseline, midline, and endline.
8.	Table 2: For the numbers for source of income, does it include those who have a job? Because the options do not reflect that	Thank you for your comment. The numbers for source of income should not include those who indicated that they had a job. However, some participants who had a job also answered this question as their job was not their primary source of income

9.	Table 3: Practices (menstrual cups) has very high Confidence Intervals for the measures of effect. What is accounting for that? And how do you interpret this?	The high confidence interval suggests that the sample size for practices (menstrual cups) was very small and thus could not provide a precise representation of the population mean. This result aligns with the study findings as a very small proportion of the cohort chose to use the menstrual cup.
10.	Discussion: You could discuss the possibility of lack of repeated interventions ..eg for the education, on the reduction in the measure of outcomes over time.	Thank you for your suggestion. Participants had access to CHIEDZA and the MH intervention throughout the follow-up period. However, they were not mandated to engage with MH services within CHIEDZA outside of their follow-up visit dates and this may have informed the results. We have revised the manuscript to reflect this consideration: “While there may be additional factors that informed the 12-month visit dip in MH knowledge, perceptions, and practices in our study, such as participants’ levels of engagement with MH activities within CHIEDZA, our results suggest the need for MH interventions that include community members and prioritise community sensitisation and education around MH-related issues if there are to be acceptable, effective, and sustainable over time.” Page 25, Lines 472 - 3
11.	Again under limitations, you have not yet shown that LTFU had no effect so please revise	Thank you for your comment. We have now revised Table 2 to reflect that LTFU had no significant effect on the study. Pages 16 - 17
Reviewer 2		

1.	The manuscript is well written. There are a few recommendations that can enhance the manuscript's quality. Regarding the cultural and religious barriers or influences for MH (pages 405-408) authors include sociocultural and environmental factors, including cultural beliefs, as factors outside the intervention's scope, that may attenuate positive effects of the intervention over time. Could the cultural and religious influences be written under the exclusion or exclusion criteria as well?	Thank you for your comments and for reviewing this paper. Based on the analysis of the quantitative and qualitative data, we believe that the sociocultural and environmental factors such as cultural and religious beliefs informed MH practices and perceptions over time. However, we cannot include these in the exclusion criteria for the cohort as these factors were not exclusive to the participants alone but also include the sociocultural beliefs of individuals and community outside our study setting e.g., participants' family members, partners, and friends. Also, these factors' influence on MH outcomes is something that became most apparent after analysing the qualitative data.
2.	Page 7 Lines 147-149 Regarding the fact that the study was nested within a cluster randomized trial of community-based integrated HIV and SRH intervention for young people aged 16-24 years (CHIEDZA); this information should also be reported as a bias/limitation since only a select or convenience sample of females (CHIEDZA clients) who attend these sites participated	Thank you for comment. We have included this limitation in the revised manuscript: "The cohort participants were recruited in two randomly selected CHIEDZA intervention clusters (Hatcliffe and Budiro) within one province (Harare) of Zimbabwe. Therefore, the relatively small number of participants in the cohort came from similar backgrounds and have similar characteristics making the study results less generalizable." Page 26, Lines 491-4
3.	Pages 19-22, under results, it may be redundant to use quotes in the main manuscript as well presenting the exact quotes as Table 4 on page 23.	Thank you for your comment. We agree and have removed the repeated quotes from Table 4 in the manuscript to avoid repetition.
Editor comments		
1.	Please note that declarative titles are not part of the journal format. As such, please revise the title of your manuscript to include the research question, study design and setting. This is the preferred format of the journal. See published articles for examples.	Thank you for your comments on this manuscript. We have revised the manuscript title to include the research question, study design, and setting:

		“A mixed-methods study measuring the effectiveness of menstrual health intervention on menstrual health knowledge, perceptions, and practices among young women in Zimbabwe” Page 1, Lines 2 - 3
2.	Along with your revised manuscript, please include a copy of the ENTREQ checklist, for reporting of synthesis of qualitative research, indicating the page/line numbers of your manuscript where the relevant information can be found https://bmcmmedresmethodol.biomedcentral.com/articles/10.1186/1471-2288-12-181#Tab1)	We have included a filled out copy of the ENTREQ checklist.
3.	Please provide more information about the informed consent process. For example, was parental consent required for participants under the age of 18? And if not, why was this the case?	Thank you for your comment. We have revised the manuscript to give more detail on the informed consent process and national guidelines: “Written informed consent was obtained from all participants in the cohort study. No parental consent was needed to uptake the MH intervention as national guidelines allow for those aged 16 years and older to give independent consent to accessing SRH services.” Page 9, Lines 188 – 190 “Written informed consent was obtained before the FGDs were initiated and pseudonyms were used throughout for confidentiality and maintain anonymity. National guidelines indicate that parental consent is required for those under 18 years old to participate in research but due to the risk of desirability bias affecting responses and the minimal risk associated with participation in FGDs and IDIs, a waiver for parental consent for participants 16–18 years old was obtained from the ethics review bodies. FGDs were audio recorded, transcribed verbatim, and then translated into English where necessary.”

VERSION 2 – REVIEW

REVIEWER	AMOAKOH-COLEMAN, MARY University of Ghana
REVIEW RETURNED	07-Jan-2023
GENERAL COMMENTS	Good to go